# The Unmet Needs for Studying Chronic Pelvic/Visceral Pain Using Animal Models

**DOI:** 10.3390/biomedicines11030696

**Published:** 2023-02-24

**Authors:** Ana Catarina Neto, Mariana Santos-Pereira, Pedro Abreu-Mendes, Delminda Neves, Henrique Almeida, Francisco Cruz, Ana Charrua

**Affiliations:** 1Experimental Biology Unit, Department of Biomedicine, Faculty of Medicine of University of Porto, 4200-319 Porto, Portugal; 2I3S—Instituto de Investigação e Inovação em Saúde, University of Porto, 4200-135 Porto, Portugal; 3Department of Urology, Centro Hospitalar de São João, 4200-319 Porto, Portugal; 4Physiology and Surgery Department, Faculty of Medicine of University of Porto, 4200-319 Porto, Portugal; 5Ginecologia-Obstetrícia, Hospital-CUF Porto, 4100-180 Porto, Portugal

**Keywords:** endometriosis pain syndrome, chronic primary bladder pain syndrome, animal model

## Abstract

The different definitions of chronic pelvic/visceral pain used by international societies have changed over the years. These differences have a great impact on the way researchers study chronic pelvic/visceral pain. Recently, the role of systemic changes, including the role of the central nervous system, in the perpetuation and chronification of pelvic/visceral pain has gained weight. Consequently, researchers are using animal models that resemble those systemic changes rather than using models that are organ- or tissue-specific. In this review, we discuss the advantages and disadvantages of using bladder-centric and systemic models, enumerating some of the central nervous system changes and pain-related behaviors occurring in each model. We also present some drawbacks when using animal models and pain-related behavior tests and raise questions about possible, yet to be demonstrated, investigator-related bias. We also suggest new approaches to study chronic pelvic/visceral pain by refining existing animal models or using new ones.

## 1. The Implications of the Definitions and Terminologies of Chronic Pelvic and Visceral Pain

According to the International Association for the Study of Pain (IASP), chronic primary pain is defined as a persistent pain lasting for 3 months that is associated with emotional and/or functional impairment and cannot be associated with another condition [1]. If pain is associated with the head/neck viscera, thoracic viscera, abdominal visceral, or pelvic viscera with referral pain patterns from specific visceral organs, it is called chronic primary visceral pain (CPVP) [1]. If pain is felt in the pelvic region with referral pain patterns from specific pelvic organs without a diagnosed cause, after excluding all other causes it is called chronic primary pelvic pain syndrome (CPPPS) [1]. Endometriosis pain syndrome (EPS) is an example of CPPPS and should not be confused with endometriosis, whose lower abdominal pain is related to ectopic endometrium and, therefore, considered a chronic secondary visceral pain syndrome caused by persistent inflammation/vascular mechanisms (Table 1) [1,2,3]. However, the American Association of Gynecologic Laparoscopists, the European Society for Gynecological Endoscopy, the European Society of Human Reproduction and Embryology, and the World Endometriosis Society do not distinguish the chronic pain associated with EPS and endometriosis, assembling the patients under an umbrella term of endometriosis [4,5]. According to the IASP, if the pain is felt in the urinary bladder and there is at least one lower urinary tract symptom, without apparent cause or bladder inflammation, it is called chronic primary bladder pain syndrome (CPBPS), formerly known as interstitial cystitis, painful bladder syndrome, and PBS/IC or BPS/IC (Table 1) [1]. The 2022 European Association of Urology Guidelines on Chronic Pelvic Pain followed the IASP definitions and nomenclature [6]. The 2022 American Urology Association Guidelines did not approach the chronic pelvic/visceral pain terminology issue [7].

According to the International Continence Society (ICS) definitions, chronic pelvic pain (CPP) is a persistent, recurrent, or continuous pain for 6 months that is related to the pelvic/abdominal region and is often associated with gynecological and lower urinary tract symptoms. If central sensitization and other chronic pain mechanisms are identified, the definition of chronic pelvic pain also applies, even if the general rule of six months is yet to be confirmed [8]. According to the ICS, whenever chronic pelvic pain cannot be associated with a well-defined pathology, it is called chronic pelvic pain syndrome (CPPS) [9]. Bladder pain syndrome (BPS) is an example of CPPS (Table 1). The European Society for the Study of IC/BPS (ESSIC) follows the same definition as the ICS [10]. According to the ICS, if chronic pain is associated with visceral organ damage, inflammation, or pressure and is detected by visceral organ nociceptors, it is called chronic visceral pain (CVP) [8]. Endometriosis is an example of CVP with an organic cause, specifically with an inflammatory origin (Table 1) [8]. The ICS does not have a definition for EPS.

**Table 1 biomedicines-11-00696-t001:** Summary of IASP and ICS classifications of pain related to endometriosis, endometriosis pain syndrome (EPS), and bladder pain syndrome (BPS) and chronic primary bladder pain syndrome (CPBPS).

	IASP/EAU	ICS/ESSIC
Endometriosis	Chronic secondary visceral pain syndrome	Chronic visceral pain
EPS	Chronic primary visceral pain—chronic primary pelvic pain syndrome	-
CPBPS/BPS	Chronic primary visceral pain	Chronic pelvic pain
Interstitial cystitis	-	Chronic pelvic pain

Both the IASP/EAU and ICS/ESSIC classifications consider that endometriosis is associated with visceral pain. However, while in the IASP/EAU definition pain is due to the possible extra-pelvic location of the lesion [1,2,6] and the term secondary indicates that it is associated with tissue inflammation, in the ICS/ESSIC definition pain results from the tissue inflammation detected by nociceptors [9].

The ICS/ESSIC do not have a definition for EPS. The ICS classification is based on symptoms, and therefore it may not distinguish endometriosis from EPS [9]. Hence, endometriosis may have been used to define EPS in many clinical studies and reports. Accordingly, the ICS/ESSIC pain classification also does not distinguish bladder pain syndrome from interstitial cystitis, as patients may present similar pain symptoms, although it approaches them as two different conditions. The evolution of the bladder pain syndrome nomenclature showed that in the past, interstitial cystitis was used to define bladder pain syndrome in clinical studies and reports.

Although EPS patients may have extra-pelvic lesions and/or may report extra-pelvic pain, the IASP/EAU define it as pelvic pain. On the other hand, CPBPS is not considered a pelvic pain syndrome by the IASP/EAU, probably because the location of pain, although difficult to identify, is mostly (but not exclusively [11]) found in the visceral region. In the ICS/ESSIC definition, however, the pelvic classification is due to the lack of connection of the pain to a visceral organ impairment [9].

The different criteria used by the IASP/EAU (origin and location) and the ICS/ESSIC (functional/symptoms) definitions of pain related to endometriosis and bladder pain syndrome influence the approaches to study these conditions. While using the IASP/EAU classification, the three identified conditions are endometriosis, EPS, and CPBPS (without IC). Using the ICS/ESSIC classification, the three identified conditions are endometriosis, BPS, and IC.

## 2. The Implications of Symptom Identification in the Study of Chronic Pelvic/Visceral Pain

One of the characteristics of EPS is that it was previously diagnosed as endometriosis, which justifies the origin of the pelvic pain [1,2]. However, after the surgical removal of the ectopic endometrium or its hormonal treatment, the pain may remain [1]. Therefore, the existence of ectopic endometrium in EPS patients may be, in some cases, irrelevant to the pain phenotype [1]. CPBPS, on the other hand, does not require the existence of inflammation and tissue damage in its diagnostic, although some patients might present inflammatory cells in the urinary bladder wall [1].

One approach to understanding if the mechanisms of chronic pain associated with EPS and CPBPS overlap is to identify the common features of these conditions. One common aspect is the location of pain described by patients. Most of the work published concerning the mapping of pain in endometriosis was performed pre-surgery and complemented by post-surgery diagnosis. Hence, it is not clear which pain locations were related to endometriosis and which were related to EPS or even if locations were related to EPS and changed after the removal of the lesions. Nevertheless, there was an extensive overlap in the mapping of the pain of endometriosis and CPBPS, as shown in Figure 1.

**Figure 1 biomedicines-11-00696-f001:**
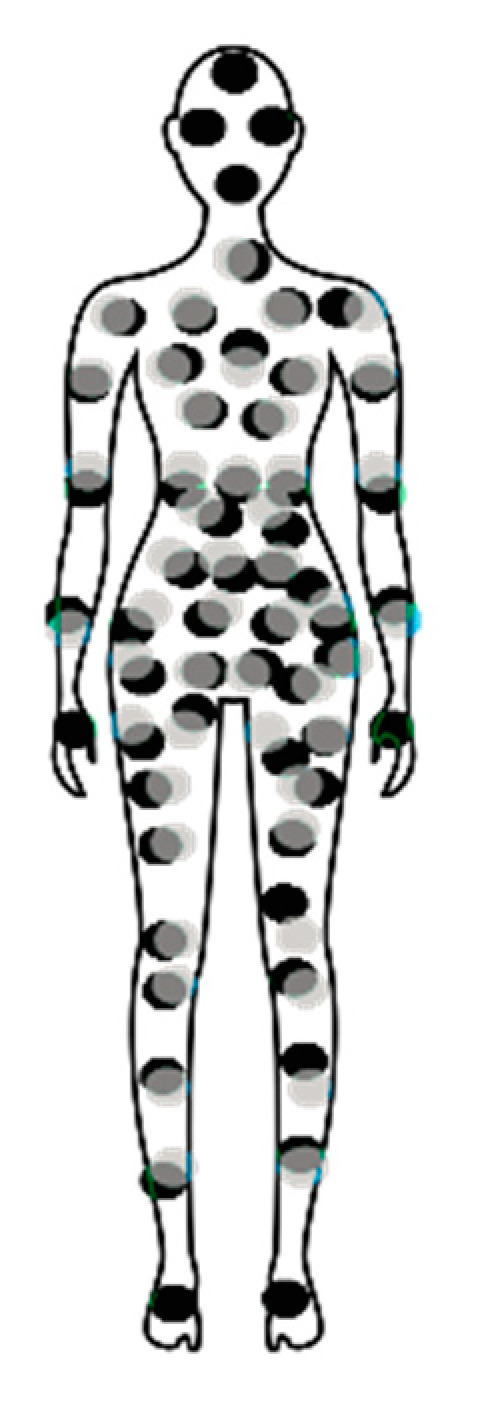
Illustrative image of pain mapping in women with endometriosis (grey dots, adapted from [12]) and CPBPS (black dots, adapted from [11]), regardless of the reported intensity of the pain.

Another common aspect in both syndromes is that patients present cognitive, behavioral, sexual, and/or emotional modifications, which are reflected in comorbidities such as depression and anxiety [1,13,14,15]. Moreover, the same patient may be diagnosed simultaneously with EPS and CPBPS [16]. It is not surprising, therefore, that both syndromes are associated with changes in the central nervous system [16], with the insula and the thalamus being the most commonly altered brain structures in both syndromes [17,18,19,20]. It is also noteworthy that both EPS and CPBPS present changes in the hypothalamic–pituitary–adrenal (HPA) axis, altering the stress-mediated response to noxious stimuli [16,21,22]. In fact, changes in cortisol levels have been associated with pain intensity (negative correlation) in these conditions [22,23]. Concurrent with HPA-axis hormonal changes, patients with both syndromes simultaneously present changes in the autonomic nervous system [24,25]. Therefore, both EPS and CPBPS patients present several phenotypes, which may explain why the available therapies are not pan-effective. As a corollary, many clinicians strongly support clustering patients by clinical and histological findings and by their responses to specific therapies or even to individualized management.

## 3. Back-translation of Patient Symptoms According to Their Associations with Chronic Pelvic/Visceral Pain

The clustering of patients according to their symptoms and signs is a complex issue. The multiplicity of phenotypes presented by patients and the multiplicity of scientific organizations make the definition of a worldwide accepted cluster very complex. Nevertheless, some attempts have been made. The UPOINT system proposed by Nickel and co-workers for phenotyping CPBPS patients is a good example of a simple tool to phenotype patients [26]. In the UPOINT algorithm, symptoms and signs are not exclusively related to the urinary bladder [26,27]. Rather, UPOINT is an acronym for the different domains observed in or reported by patients, which include urinary symptoms, psychosocial dysfunction, organ-specific findings, infection, neurologic/systemic comorbidities, and tenderness (related to pain) [26]. By broadening the urinary symptoms concept into functional symptoms, UPOINT may help to cluster patients with EPS, CPBPS, and even other chronic pelvic/visceral pain syndromes. The UPOINT score has shown strong positive correlations with the score obtained using the Interstitial Cystitis Symptom Index (ICSI) questionnaire and with the pain score obtained using a visual analogue scale (VAS) [26,27]. UPOINT has evolved to include other symptoms, including sexual dysfunction [28], and has already been used to define the therapeutic approach according to the patient’s phenotype [29,30]. That is, the presence and severity of the symptoms observed in each subdomain of UPOINT determine if a single or a multimodal therapy should be offered to the patient [29].

Back-translation of the UPOINT may help researchers to choose and refine the animal models used to study chronic visceral/pelvic pain. Some aspects of EPS and CPBPS are more easily back-translated than others, either due to methodological limitations or due to the pain characteristics of individual subjects. Taking the organ-specific findings as an example, in EPS one of the most common organ-specific findings is the presence of endometrial lesions. Nonetheless, the number of lesions in patients does not correlate with the intensity of pain [31]. Moreover, the presence of lesions in patients is not a necessary condition for the development of pain [32,33]. This is an important concept, as back-translated data are generally supported by clinical studies that use patients with chronic pain without endometrial lesions as controls for patients that present endometrial lesions.

Recent works have suggested that, independent of being related to endometriotic lesions, the pain associated with EPS results from changes in the way it is integrated in the central nervous system (CNS) [32]. Hence, when using the induction of lesions in an EPS animal model, those lesions should induce changes in the same way that the patients’ CNSs integrate pain. Moreover, to better replicate what is observed in EPS patients, the CNS changes should persist after the removal of those lesions (as occurs during EPS) to ensure that the observed animal pain behavior is not a consequence of the inflammation generated by the lesions (as occurs during endometriosis). However, this is methodologically complex, time-consuming, and, to our knowledge, has never been performed. Hence, the finding of specific markers for these CNS changes will help to surpass this methodological drawback.

In CPBPS, one of the most common organ-specific findings is urothelium functional impairment (not denudation, as in the case of IC). The urothelium barrier impairment has been associated with submucosa inflammation of the urinary bladder in patients with CPBPS and other bladder pathologies [34,35]. The urothelial barrier impairment is thought to be the mechanism by which the urine content activates bladder nociceptors that, in turn, induce neurogenic inflammation locally and promote CNS changes that lead to pain chronification [36]. This hypothesis, however, is difficult to test using bladder-centric CPBPS animal models, not only because of the technical complexity but also because of the difficulty in defining if the animal pain-like behavior is due to neurogenic inflammation (as might predominate in IC) or due to the changes induced in the CNS (as happens during CPBPS). Hence, the identification of which/how symptoms correlate with the etiology of chronic pelvic/visceral pain phenotypes is difficult in animals, especially when it is apparent that different insults/events may lead to similar chronic pelvic/visceral pain phenotypes.

## 4. Systemic Symptoms Prompt Systemic Models for the Study of Chronic Pelvic/Visceral Pain

EPS and CPBPS patients commonly present systemic symptoms, potentially reflecting the existent comorbidities and CNS changes that translate into the multiple pain sites observed in Figure 1. Hence, animal models with CNS alterations that lead to pain behaviors and/or contribute to the development of symptoms of comorbidities (changes in organ activity and/or sensitivity as well as depression and/or anxiety symptoms, among others) may potentially provide more information about the chronic pain associated with EPS (white, Table 2) and CPBPS (grey, Table 2) than organ-centered models. In Table 2, the most used complex rodent models of EPS/endometriosis and CPBPS [37] and how they relate to UPOINT are summarized.

**Table 2 biomedicines-11-00696-t002:** Rodent models used recently to study EPS (white background) and CPBPS (grey background). The animals’ characteristics are organized according to their similarity to patients’ symptoms and signs in each UPOINT subdomain.

Model	U	P	O	I	N	T	References
Retrograde menstruation						+	[38]
Autologous transplantation	**+**	**+**	+		**+**	+	[39,40,41,42,43,44,45,46,47,48,49,50,51,52,53,54,55,56,57,58,59,60,61,62,63]
Allogenic transplantation						+	[39]
Autologous transplantation to the sciatic nerve					**+**	+	[64,65,66,67,68,69,70]
Autologous transplantation to muscle						+	[71,72,73,74,75]
Syngeneic transplantation	**+**		+		**+**	+	[76,77]
Xenograft transplantation						+	[78]
Intraperitoneal injection of endometrial tissue						+	[79,80,81]
Peritonitis					**+**	+	[82,83]
Colonic instillation of TNBS	**+**		+		**+**	+	[84,85,86,87,88,89]
Uterine pain					**+**		[90]
Autoimmune	**+**		+		**+**	+	[91,92,93,94,95,96]
Prostatic inflammation	**+**			+	**+**		[97,98,99,100]
Pseudorabies virus			+	+	**+**	+	[101,102,103,104,105,106,107,108]
Systemic administration of molecules	**+**				**+**	+	[109,110,111]
Restraint stress	**+**		+		**+**		[112,113,114,115]
Water avoidance stress	**+**	**+**	+		**+**	+	[116,117,118,119,120,121]
Chronic variable stress	**+**		+				[122]
Cold stress	**+**						[123,124]
Social stress	**+**		+		**+**		[125,126,127]
Foot shock stress		**+**			**+**	+	[128,129]
Early-life stress by odor–shock conditioning			+		**+**	+	[130,131]
Early-life stress by neonatal maternal deprivation	**+**	**+**	+		**+**		[132,133,134]

Among the EPS/endometriosis and CPBPS/BPS/IC models in use, those that more similarly back-translate the systemic symptoms are those that present urinary/functional symptoms, psychosocial dysfunction, and neurologic/systemic (comorbidities) impairments. This is the case of the autologous transplantation, water avoidance stress (WAS), and neonatal maternal deprivation (NMD) models. However, even these models need further refinements if the goal is to unveil the pathophysiological mechanism or find new treatments for EPS and CPBPS.

Considering the autologous transplantation model (ATM), endometrial tissue transplantation must be performed because the rodents do not have a menstrual cycle and consequently do not spontaneously develop endometrial lesions. The only exception is the spiny mouse [135]. Nevertheless, as in any other rodent species, the spiny mouse also possesses an ovarian bursa that encapsulates the ovaries and the oviducts, not allowing the endometrium tissue to escape to the peritoneal cavity. Only recently, an ovarian endometriosis model was developed in spiny mice via bursectomy [136], which provides a more physiologic endometrial peritoneal lesion model for investigation. The most used ATM, normally induced in BALB/c mice, offers the possibility, within its limitations, to study the chronic pain associated with endometriosis once the transplantation is performed and the ectopic lesions are formed [135]. However, as stated before, the pain induced in this model may be of inflammatory/neurogenic origin and may not reflect the CNS changes observed in patients with EPS. In fact, variations (species and lesion location, among others) in ATM model induction may be enough to induce different modulations of the brain activity, which are then reflected in distinct pain-related behavior. For instance, the use of the ATM in Sprague-Dawley rats (implantation in the peritoneal wall) induces increased regional homogeneity of the anterior cingulate cortex, in the CA1 field of the hippocampus, in the thalamus, and in the brainstem, showing that these regions have abnormal neuronal activity [137]. A high number of neuronal cells show the activation of the apoptosis cascade [137]. Apart from the thalamus, the overactive neurons found in these nuclei show higher TRPV1 mRNA levels [137]. In addition, the overactive neurons from the hippocampus present increased levels of NMDA receptor mRNA [137]. On the other hand, there is a decrease in the regional homogeneity of the basomedial nucleus of the amygdala and in the primary motor cortex [137], providing evidence of altered basal brain activity. All these changes are concomitant with the development of thermal hyperalgesia and mechanical allodynia but not anxious behavior [137]. The ATM induced in C57BL/6 mice (implantation in the peritoneal wall) presents some similarities to the induction of the ATM in Sprague-Dawley rats (such as changes in hippocampus activity) but presents other important differences, such as changes in the activity of different brain areas, namely the insula [138]. Moreover, the observed changes in neuronal activity in the amygdala may differ, as in C57BL/6 mice, considering that the ATM increased the expression of the emotional regulation gene Lcn2, which may explain the increased anxiety induced by the ATM in these animals [138]. From a behavior phenotype point of view, the ATM induces thermal hyperalgesia in C57BL/6 mice but decreases locomotor activity [138]. On the other hand, this animal model presents changes in the thalamus and anterior cingulate cortex that are concomitant with anxiety behavior and mechanical allodynia [77].

Another model is the water avoidance stress (WAS) model. In this model, pain induction involves the activation of the peripheral adrenergic system [116]. How long does the pain phenotype induced by stress last? WAS carried out for 10 consecutive days was shown to induce a pain-related visceromotor response that lasted, in some animals, for 40 days but induced somatic nociception, which faded away much sooner. Is a 10-day WAS protocol sufficient to make an imprint in the CNS? When studying chronic pelvic/visceral pain using animal models of EPS and CPBPS, CNS pain integration changes may need more time to occur/establish than the other symptoms commonly associated with these syndromes. The great advantage of the neonatal maternal deprivation (NMD) model is that the pain-related phenotype occurs and is studied in adulthood, when the early-life insult is long gone. However, NMD is not free of drawbacks. The spontaneous pain behavior may not be easily observed, although the changes in the CNS pain integration pathway are already provoked.

Which test should be used to study chronic pelvic/visceral pain? What endpoint should be looked for? Many reviews have brought to light the advantages and disadvantages of using spontaneous and induced pain behavior tests [139,140]. The induced pain behavior tests, such as von Frey filament stimulation or the hot plate test, are normally based on the reflexive and conscious withdrawal of body parts submitted to a stimulus [139]. Therefore, their principal endpoints, based on nocifensive behaviors, are normally the latency time to a response or equivalent to an LD50 threshold [139]. Although these tests are indisputably associated with a bothersome/painful sensation, the focal location of such a sensation is hard to determine in the case of chronic pelvic/visceral pain. Additionally, they do not allow the spontaneous pain felt by the animals to be measured. Hence, it is imperative to find adjusted endpoints to study chronic pelvic/visceral pain that may overcome the drawbacks of the classically used tests, including the concomitant use of behavior tests with other techniques, such as neuroimaging or functional/electrophysiological measurements. In fact, tests that are based on pain-related spontaneous and/or cognitive behaviors (psychosocial dysfunction and the neurologic/systemic aspect of the UPOINT approach) associated with animal quality of life have been used more recently [139]. However, such tests are more susceptible to some pitfalls that are not directly associated with the tests per se, which may endanger their interlaboratory comparison and impair their validity. These pitfalls are related to the animals’ species-specific characteristics, animal husbandry and manipulation conditions, and/or data acquisition requirements, which are often overlooked and therefore not included in standard operating protocols. Additionally, the use of social cues also raises the question of the impact of genetic/epigenetics, which are known to be implicated in the balance of the HPA axis, in the study of chronic pelvic/visceral pain. The induction of WAS for 7 days is known to induce changes in the expression of glucocorticoid receptor and corticotropin-releasing factor genes of neuronal cells of the amygdala of male F-344 rats that are intimately related with increases in pain-related visceromotor responses through epigenetic mechanisms [141]. Therefore, we should evaluate if, for example, outliers should be disregarded. Moreover, investigators should not expect all animals to develop pain phenotypes after the induction in each experimental model, a piece of evidence that raises the question of how to handle the groups. Should we use different strains in the same protocol? What is the real impact of litter balancing in the study of chronic pelvic/visceral pain?

It is clear that the animal models of EPS and CPBPS require better characterization before deciding which tests should be used in the evaluation of the pathophysiological mechanisms of chronic pain or even pain treatments. For NMD, it should be elucidated which symptoms and signs NMD replicates and which pathways induce the NMD pain/sensitization pathway and associated behavior changes, such as anxiety. For instance, NMD is known to increase perigenital sensitivity and increase the number of voiding episodes in adult male mice [133]. NMD also influences the sensitivity of the colon, as it increases the visceromotor response to colorectal balloon distention in adult male rats [142], a fact that may accelerate the progression of endometrial lesions in female mice [143]. NMD causes mastocytosis in the bladder and prostate in adult male mice [133,144]. NMD also promotes the impairment of hippocampal neurogenesis and alters stress-related gene expression within the hypothalamic–pituitary–adrenal axis [145]. Moreover, it was recently postulated that NMD may prime a glial response that is reflected in adulthood in the development of pelvic/visceral pain, similar to what is observed in somatic pain [146].

Besides the rodent models, non-human primates (NHP) and domestic cats have been used to study EPS and CPBPS, respectively [147,148,149,150]. NHPs share similar reproductive system anatomy and endocrinology with humans. Baboons and some macaca species (such as rhesus, pigtail, and cynomolgus) menstruate and may develop spontaneous endometriosis [151,152,153]. The EPS in macaques is very similar to the condition in humans, as they present pain behavior, and the functional magnetic resonance imaging of their brains revealed a sensitized insula and thalamus [154]. Moreover, their endometrium and endometriosis tissues present mitochondrial impairment, which may be related to oxidative stress [155], a feature also observed in patients. As happens with some endometriosis patients, some macaques are anemic [156]. Furthermore, it is possible to find other species that spontaneously develop endometriosis in non-menstruating animals such as gorillas, horses, dogs, and guinea pigs, among others [157,158,159]. Despite the straightforward phylogenetic gap between humans, rodents are, at the end of the day, animal models with a high level of physiological homology. The ethical problems associated with the use of many other species, the maintenance expenses, and the decreased number of animals available for experimentation when compared to rodents make non-rodent models difficult to use routinely. The same holds true for CPBPS. Cats may develop feline interstitial cystitis (FIC), which is a similar form of CPBPS [160]. These cats exhibit flare-like symptoms, lower urinary tract abnormalities, and central nervous changes, and in some cases they present comorbidities such as behavioral, endocrine, cardiovascular, and gastrointestinal alterations [150]. However, the low availability of cats with FIC and the high cost necessary for their maintenance when compared with rodents make this FIC model less attractive [160,161].

## 5. Going beyond the Existing Models to Study the Chronic Pelvic/Visceral Pain Associated with EPS and CPBPS

To study EPS, the ATM can be combined with procedures/protocols that alter the way the CNS processes pain (Figure 2). For instance, the combination of the ATM with stress models (known to alter the HPA axis and/or induce CNS changes [141,162]) or with autoimmune models (known to induce CNS sensitization due to prolonged inflammation and/or tissue damage [163]) may help to refine the ATM to study EPS. In fact, female Balb/C mice submitted to NMD presented depression-like and anxiety-like symptoms in adulthood [143]. However, it was only after the induction of endometrial lesions in these animals that it was possible to observe endometriosis-associated generalized hyperalgesia [143]. Another putative approach is the use of longer ATM protocols to understand if, at longer time points, the CNS changes that lead to chronic pain are already observed.

**Figure 2 biomedicines-11-00696-f002:**
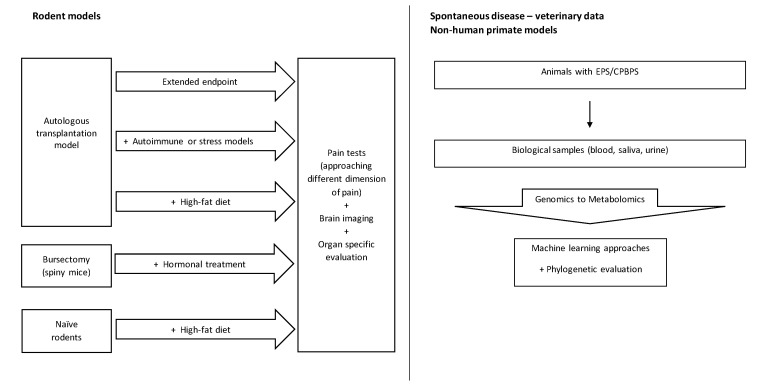
Possible refinement strategies to improve the study of chronic pelvic/visceral pain using animal models or animals that naturally develop EPS or CPBPS.

A frequently disregarded aspect regarding chronic pelvic/visceral pain is the role of visceral fat. EPS/endometriosis are related to an imbalance in the levels of estrogen, a hormone known to modulate the distribution and amount of fat tissue in the body [164,165]. Moreover, the lesions commonly observed in EPS/endometriosis occur in close proximity to the visceral fat present in the pelvic cavity [166,167,168,169]. The visceral fat is known to secrete pro-inflammatory molecules, which may promote tissue inflammation, even in the absence of endometrial lesions. In a study involving 3026 patients with osteoarthritis, visceral fat was associated with an increased pain phenotype, regardless of BMI [170], showing that the systemic inflammation induced by visceral fat promotes changes in the pain phenotype. In a study with 795 participants, it was observed that visceral fat was associated with sympathetic system activity [171]. Accordingly, visceral fat innervation is known to develop crosstalk with brain centers, namely the paraventricular nucleus (PVN) of the hypothalamus [172]. The hyperactivation of the PVN nucleus, as occurs in stressful situations, is thought to have a role in endometriosis/EPS, as it promotes HPA axis activation, modulating cortisol release concomitantly with endometrial dysfunction [170,173]. Therefore, combining a high-fat diet with lesion induction might provide a valid approach for studying EPS (Figure 2), as shown by Heard and colleagues in 2016 [174]. Moreover, changes in PVN activity, namely the increased release of corticotropin-releasing hormone (CRH), might also have a role in CPBPS, as these patients present urothelial expression changes in those hormone receptors [175]. The change in CRH receptors is intimately related to pain scores and markers in those patients [175]. Hence, high-fat-diet animal models may provide the chance to study features that are overlapped in EPS and CPBPS.

Besides the induction of chronic pelvic pain in animal models, a more exhaustive study of biological samples (ectopic endometrial tissue, blood, saliva, and urine) taken from animals that naturally develop EPS or CPBPS, together with other veterinary analyses that might be performed on those animals, should be considered. Important lessons can be taken, such as the possibility of performing a phylogenetic evaluation of the disease. An example of this is the recent genetic comparison between humans and the rhesus macaque that unveiled the importance of the coding variants in neuropeptide S receptor 1 in the pain treatment associated with endometriosis [158]. The phylogenetic evaluation of the disease may provide important clues based not only on the similarities but also on the most noticeable differences. It should be recalled that FIC affects both genders in a relatively equal way [150,160], a characteristic that may help investigators to understand CPBPS in the male population.

## 6. Conclusions

Altogether, the refinement of the translational chronic pelvic/visceral pain model to better understand the similarities with human conditions is essential to perform more meaningful studies. In addition, the study of specific cellular and subcellular mechanisms in the peripheral organs and the central nervous system nuclei will be fundamental to understanding the pathophysiology of EPS and CPBPS and to developing more effective treatments in a systemic setting.

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
