# Peer review of "The Unmet Needs for Studying Chronic Pelvic/Visceral Pain Using Animal Models"

_biomedicines, 2023, doi:10.3390/biomedicines11030696_

Round 1

Reviewer 1 Report (Previous Reviewer 2)

The authors present a narrative review of a very scarce innovation and with discreet innovation. The manuscript is written erratically and with inadequate references. The authors fail to discuss the novelty and the scope of this manuscript is inadequate. This study looks more like a popularization manuscript.

Author Response

We thank the reviewer´s comment.

Endometriosis is a chronic disease that provokes pelvic pain. Actually, this is the main symptom of endometriosis patients. Endometriosis-associated pelvic pain is not easy to control, considering that it tends to be refractory to the common treatments, such as analgesics and hormone-based therapies. In addition, neuropathic hyperalgesia in often observed in endometriosis patients. Taking this evidence into consideration, we developed a review on the topic of animal models of chronic pelvic pain. We consider that this subject is of major relevance in the setting of endometriosis research and under the scope of the Biomedicines special issue “Animal Models of Endometriosis, from the Bench to the Clinic”. As far as we know, no equivalent review that aggregates what is known about pelvic pain on the setting of inflammatory diseases, is published. This is the novelty of this review.

To prepare this review we proceeded to an accurate analysis of the published studies employing animal models of pelvic pain and all those articles are properly referenced in the text. To meet the reviewer suggestions, we have revised all the references and changed them when appropriated.

The study of chronic pelvic/visceral pain is being performed by a wide range of specialist coming from the most diverse fields, going from medical doctors to computational scientist. Moreover, this subject is also of interest to non-scientific community, including patients organizations. Hence, now more than ever, simple but accurate language should be used to address complex concepts as this is and to make a topic comprehensible to a vast audience.

Reviewer 2 Report (New Reviewer)

Interesting article, well written and designed .

A question what is the primary and most 

Essential endpoint?

The quality of presentation is good

Author Response

We thank the reviewer´s comment. In our opinion, there is not a primary most essential endpoint as is not possible to analyse all the dimensions of pain in an animal model. The impossibility of communication between the animal and the handler/research does not allow the later to understand the degree of pain that the animal might be feeling. As stated in the text “Hence, it is imperative to find new endpoints to study chronic pelvic/visceral pain, that may overcome the drawbacks of the classically used tests, including the concomitant use of behaviour tests with other techniques, like neuroimaging or functional/electrophysiological measurements.”, the use of different approaches that analyse the different dimensions of pain will help to mitigate the drawbacks of each approach per se.

Reviewer 3 Report (New Reviewer)

I think this is a very extensive and thorough review from Neto and Pereira et al describing the needs for studying chronic pelvic, visceral pain using animal models. The review design is thoughtful and well-performed, however, I would like to see a schematic representation as a figure that would show a summary of the model systems and strategies that the authors envision would improve the therapeutically improve chronic pelvic and visceral pain. With that this study would be suitable to be published in MDPI-Biomedicines.

Author Response

We thank the reviewer´s comment. We have added a new figure 2 with the summary of models/strategies to refine the existing models and to create new models that will putatively improve the study of chronic pelvic/visceral pain and, therefore, may contribute to finding new therapeutic alternatives.

Round 2

Reviewer 1 Report (Previous Reviewer 2)

The authors have submitted the revised version of the manuscript entitled: The unmet needs for studying chronic pelvic/visceral pain using animal models.

The narrative review of the literature fails to present an update on a topic of high importance. The authors have not improved the manuscript in any way. This narrative review fails to meet adequate quality standards for this Journal.

This manuscript is a resubmission of an earlier submission. The following is a list of the peer review reports and author responses from that submission.

Round 1

Reviewer 1 Report

In this manuscript, the authors described the use of animal models for the study of ENDO and BPS. They reviewed studies to conclude on how representative the results are for understanding the pathophysiology of these two conditions. Overall, the papers is well written and the studies well presented.  

line 33 : "But, at THE end .."

Reviewer 2 Report

Net et al. present an interesting revision in their conception. However, this manuscript does not add novelty to what exists in the literature. The authors fail to adequately and effectively bring together the information necessary for this type of review to have an impact. These types of reviews are granted as systematic reviews. To carry out a literary review of this topic, much more information is needed and provide data that readers need (therapies, clinical trials ......).

The manuscript fails to attract the reader, the title is simple and the summary is very minimal.

The manuscript is not properly written, the use of English grammar is inadequate. This manuscript is not suitable for a Journal Q1.